# Nano Ag/PPy Biocomposites Based on Graphene Oxide Modified Bacterial Cellulose from the Juice of Xinhui Citrus and Its Antibacterial Activity

**DOI:** 10.3390/mi14101809

**Published:** 2023-09-22

**Authors:** Yihong Yang, Rong Zhou, Miaoshan Yuan, Huaiwen He

**Affiliations:** School of Materials Science and Food Engineering, Zhongshan Institute, University of Electronic Science and Technology of China, Zhongshan 528400, China; zhour087470@163.com (R.Z.); yuan013652@126.com (M.Y.); hehw3@mail2.sysu.edu.cn (H.H.)

**Keywords:** bacterial cellulose, antibacterial activity, polypyrrole, silver nanoparticles, biocomposites

## Abstract

Bacterial cellulose (BC) is a green, natural biopolymer with excellent biocompatibility and a film-forming ability. However, its lack of inherent antibacterial activity restricts its application in medical materials and food preservation. In this study, BC derived from the juice of discarded Xinhui citrus was obtained through fermentation and further modified in situ with graphene oxide (GO) to obtain BC(GO). Subsequently, BC(GO) was loaded with cell-compatible polypyrrole (PPy) and antibacterial agent silver nanoparticles (AgNPs) to prepare Ag-PPy/BC(GO) composite films. Composite films were characterized using Fourier-transform infrared spectroscopy (FTIR) and scanning electron microscopy-energy-dispersive X-ray spectroscopy (SEM-EDS) to evaluate their chemical structure and morphology. The results demonstrate effective adsorption of PPy and AgNPs onto the surface of BC nanofibers modified with GO. Antibacterial experiments reveal synergistic antibacterial effects of PPy and AgNPs. The Ag-PPy/BC(GO) membranes exhibit strong antibacterial activity against both *Escherichia coli* (*E. coli*) and *Staphylococcus aureus* (*S. aureus*), with 48-h growth inhibition rates of 75–84% and 82–84%, respectively.

## 1. Introduction

Bacterial cellulose (BC) is a sustainable biopolymer that can be obtained through microbial fermentation from waste or inexpensive raw materials [1,2]. It can be modified using various chemical and physical methods to impart antimicrobial [3,4], conductive [5,6], and other multifunctional and tunable properties [7]. Modified bacterial cellulose holds significant potential and significance in antibacterial applications [8,9]. Due to its natural biocompatibility and tunability, it is an ideal antibacterial material [10]. Bacterial celluloses modified with antimicrobial agents, nanoparticles, or functional groups enhance their antimicrobial activity against a wide range of pathogens [11,12,13]. These materials exhibit extensive applications in medical devices [8,14,15], biosafety [16], and food preservation [17,18].

To enhance the antibacterial properties of BC, the incorporation of silver nanoparticles (AgNPs) has been widely explored. AgNPs are recognized as broad-spectrum antimicrobial agents with persistent inhibitory effects on microorganisms [19]. Silver nanoparticles (AgNPs) are recognized as broad-spectrum antimicrobial agents that possess persistent inhibitory effects on various microorganisms which demonstrate good adhesion, toughness, and cohesion [19]. Shen et al. introduced AgNPs into a polyvinyl alcohol/bacterial cellulose (PVA/BC) solution and prepared an excellent PVA/BC/Ag hydrogel using freeze-thawing [4]. Experimental evidence demonstrated the effective wound healing and high wound closure rate achieved by this hydrogel within a short period. Li Yi-Tsen et al. incorporated AgNPs into a montmorillonite-BC composite film to achieve antimicrobial activity and controlled release [3]. Zhang et al. coated polyvinyl alcohol and BC onto the surface of biochar and introduced AgNPs, resulting in a composite film with antimicrobial properties and controllable release characteristics [20]. These studies highlight the significant role of AgNPs in modified BC materials and provide a promising pathway for developing antibacterial materials with enhanced properties.

Graphene oxide (GO), a precursor to graphene, possesses numerous oxygen-containing functional groups on its two-dimensional layered structure, providing space for embedding polymers and inorganic nanoparticles [21]. Polypyrrole (PPy), a conductive polymer, offers advantages such as antimicrobial properties, promotion of cell growth, and biocompatibility, with broad application prospects [22]. By dispersing nanofibrillated cellulose, polymerizing pyrrole, and incorporating AgNPs, bio-composite materials with enhanced conductivity and excellent antimicrobial effects can be prepared [23]. PPy/titanium dioxide-AgNPs-modified BC films were used for the detection and measurement of pathogenic bacterial growth [24]. The combination of PPy, AgNPs, and reduced graphene oxide demonstrated high efficiency in sterilization and degradation [25]. Utilizing cellulose nanocrystals, polypyrrole, silver nanoparticles, and nickel(III) oxide nanoparticles as co-assistants enabled the fabrication of conductive nanocomposites [6]. Despite significant progress in conductive and antimicrobial nanocomposite materials, challenges and technical issues still need to be addressed to achieve sustainable and effective applications. Introducing materials such as GO, PPy, and AgNPs into BC endows it with multifunctionality and tunability, widening its potential in antibacterial applications.

Xinhui citrus, a local specialty agricultural product in Guangdong province, China, serves as the raw material for Xinhui dried tangerine peels. However, during the preparation process, people tend to discard the pulp and only retain the orange peel. This resulted in significant environmental issues such as soil and water pollution [26]. In this study, we aim to address these environmental concerns by utilizing fermented waste Xinhui citrus flesh juice as the main raw material for the production of BC membranes. These membranes are co-cultivated with GO, forming a nano-grid structure with alternating porous regions. Additionally, we introduce ions to reduce pyrrole to PPy, and load AgNPs onto the membrane, enabling the integration of multiple functionalities in a single material. This holds great promise for developing more effective and environmentally friendly antibacterial materials. The biocomposite membrane shows potential applications in packaging and preservation, inhibiting microbial growth and ensuring hygienic packaging while extending the freshness and shelf life of diverse products. 

## 2. Materials and Methods

### 2.1. Materials

Graphene oxide powder (analytical grade) was purchased from Jiazhaoye New Materials Co., Ltd. (Suzhou, China). Pyrrole (analytical grade, 99%) was purchased from Shanghai McLean Biochemical Technology Co., Ltd. (Shanghai, China). Ferric chloride hexahydrate (reagent grade) was purchased from Tianjin Yongda Chemical Reagent Co., Ltd. (Tianjin, China). Silver nitrate (AgNO_3_, analytical grade) was purchased from Tianjin Tiangan Chemical Technology Development Co., Ltd. (Tianjin, China). Xinhui citrus was purchased from Xinhui District Tianma Production Area (Jiangmen, China). Kombucha SCOBY was purchased from Kombucha Enzyme Co., Ltd. (Zhanjiang, China).

### 2.2. Methods

#### 2.2.1. Preparation of Bacterial Cellulose

A small amount of kombucha culture was taken and inoculated into a sterile seed culture medium, which was then incubated and shaken for 48 h at 28 °C and 120 rpm/min. After incubation, a small amount of the seed culture was transferred to a basic culture medium and placed in a static incubator at 28 °C for 7 days. A thick, white bacterial cellulose (BC) membrane was formed on the surface of the culture medium. The BC membrane was removed from the culture medium and treated with a 2% (*w*/*v*) NaOH solution at 80 °C for 2 h to remove impurities and cells. The BC membrane was then thoroughly washed with water until the pH of the water reached neutral.

Seed culture medium (g/L): Dissolving 100 g of sucrose and 10 g of black tea into distilled water to make a total volume of 1000 mL. The pH was adjusted to 6.8–7.0, and the mixture was boiled and then filtered to achieve sterilization.

Fermentation Culture Medium (g/L): Weighing 10 g peptone, 5 g yeast extract, 8 g disodium hydrogen phosphate, 5 g potassium dihydrogen phosphate, and 2 g magnesium sulfate to make up to 1000 mL with Xinhui citrus juice [27] and then adjust the pH to 6.0.

Preprocessing of Xinhui citrus: Freshly peeled and seeded Xinhui citrus were juiced, followed by centrifugation to extract fresh tangerine juice. Sterilize the juice before use.

#### 2.2.2. Preparation of BC(GO)

GO was sonicated for 60 min and then irradiated with ultraviolet (UV) light three times, each time for 15 min. Add an appropriate amount of GO (0.2 mg/mL) to the fermentation culture medium at a volume ratio of 1:10 (GO: fermentation culture medium) and shake for 2 h. A suitable amount of bacterial liquid from the basal culture medium to the fermentation culture medium was transferred and then shaken for 12 h in a constant temperature incubator. Then, 10 mL of the mixture was put into a sterile six-well culture dish, and the culture dish was incubated in a biochemical incubator at 28 °C for 7 days to obtain BC(GO) membrane. The composite membrane was immersed in a 2% NaOH solution at 80 °C for 2 h and then was rinsed thoroughly with flowing deionized water until reached neutrality.

#### 2.2.3. Preparation of Ag-BC(GO), PPy-BC(GO) and Ag-PPy/BC(GO) Composite Membrane

A FeCl_3_ aqueous solution (0.1 mol/L) was prepared for the synthesis of the PPy-BC(GO) and Ag-PPy/BC(GO) composite membranes. The BC(GO) membrane was immersed in a 40 mL FeCl_3_ solution for 30 min, followed by the addition of pyrrole solution at different concentrations (5 mL/L, 10 mL/L, and 15 mL/L) and was added into the FeCl_3_ solution containing the BC(GO) membrane under magnetic stirring [28]. The progressive darkening of the solution’s color indicates the formation of PPy. FeCl_3_ acts as an oxidizing agent during the PPy formation process, effectively oxidizing the nitrogen atoms on the pyrrole monomers, resulting in the generation of positive charges, and initiating the subsequent polymerization reaction among pyrrole units. Fe^3+^ actively participates in this polymerization reaction and remains incorporated within the PPy structure. After a 60-min stirring period, the reaction was terminated, and the PPy-BC(GO) membranes underwent three rinses with deionized water to remove any residual reactants or impurities. Subsequently, the membranes were pre-frozen for 24 h and then freeze-dried at −50 °C for an additional 24 h. 

We utilized a photocatalytic reduction reaction to immobilize AgNPs onto the surface of membranes. The mechanism involves the adsorption of silver ions onto the cellulose membrane upon contact with a AgNO_3_ solution. Subsequently, under ultraviolet (UV) irradiation, specific chemical functional groups on the cellulose membrane surface, such as hydroxyl groups, act as reducing agents, progressively reducing the adsorbed silver ions into nanoscale silver particles. To introduce AgNPs onto the composite membrane, the dried BC(GO) and PPy-BC(GO) membranes were immersed in AgNO_3_ solutions of different concentrations (10 g/L, 15 g/L, and 20 g/L). The membranes were then subjected to vibration on a vibrating screen for 12 h, followed by static immersion for another 12 h. Finally, the membranes were irradiated with a 365 nm, 500 W UV lamp for 15 min and rinsed with an appropriate amount of deionized water. The membranes were air-dried at 60 °C for 12 h to obtain the Ag-BC(GO) and Ag-PPy/BC(GO) membranes. The overall schematic illustration of the preparation of Ag-PPy/BC(GO) is shown in Figure 1.

### 2.3. Characterization

The morphologies were observed via a scanning electron microscope with energy dispersive X-ray spectroscopy (SEM-EDS, S-4800, Hitachi Ltd., Tokyo, Japan). A fourier transform infrared spectroscopy (FT-IR, IRSpirit, Shimadzu Corporation, Kyoto, Japan) was used to characterize the molecular structure and chemical bonds of the samples.

Swelling performance test: the mass of the membrane was determined by immersing a film with an initial mass of M_0_ (g) in the same volume of distilled water. After 60 min, the membrane was removed, and the mass was measured as M_t_ (g). The swelling ratio of each film was calculated using the following Formula (1), taking the average of three parallel samples for each film:Swelling ratio (%) = [(M_t_ − M_0_)/M_0_] × 100%(1)
where M_0_ (g) and M_t_ (g) represent the initial dry weight and weight at time t of the sample, respectively.

Mechanical performance test: Mechanical performance tests were performed in a universal testing machine (UTM 500 model, Shenzhen Sansi Zongheng Technology Co., Ltd., Shenzhen, China). Three parallel groups of specimens were prepared for testing. The samples were cut into rectangular strips with dimensions of 30 mm × 10 mm. The parameters were set as follows: stretching speed of 10 mm/min, and the distance between the grips was 1.5 cm.

Antibacterial performance test: The antibacterial activity of the composite membranes were tested on *E. coli* (ATCC 25922) and *S. aureus* (NCTC 6571). The bacterial strain was inoculated into 100 mL of sterile LB liquid culture medium and incubated at a constant temperature with shaking for 12 h (conditions: 37 °C, 180 rpm/min). The bacterial density was determined and adjusted to 10^6^ cells/mL using a multifunctional microplate reader (VICTOR^®^Nivo, PerkinElmer Instruments Co., Ltd., Waltham, MA, USA). 

A volume of 10 mL of bacterial suspension was mixed with 10 mL of blank LB culture medium in a centrifuge tube, and the same mass of the sample membrane was immersed in the mixture. The centrifuge tube was placed in a constant temperature shaker for incubation. The initial incubation time was recorded as 1.5 h. After incubation for 3 h, 4 h, 6 h, 8 h, 18 h, 20 h, 24 h, 26 h, 28 h, 30 h, 32 h, 42 h, 44 h, 46 h, and 48 h, the centrifuge tube was taken out. The optical density (OD) of *E. coli* and *S. aureus* at 500 nm was measured [29,30]. The average value was calculated using three parallel samples for each sample. The LB culture medium without bacterial suspension was used as the blank value. A control sample containing the bacterial suspension without the membranes was tested as cell growth control. BC(GO) membrane sample without antibacterial particles was used as the BC(GO) control. The growth curves of *E. coli* and *S. aureus* were plotted based on the relationship between time (t) and OD values.

The antibacterial rate of the different concentrations of antibacterial composite membranes against *E. coli* and *S. aureus* were calculated using the following Formula (2) [31]:Antibacterial rate (%) = (OD_c_ − OD_S_)/(OD_c_ − OD_SB_) × 100%(2)
where OD_s_ is the absorbance value of the sample measurement well, OD_c_ is the absorbance value of the cell growth control, and OD_SB_ is the absorbance value of the culture medium blank.

## 3. Results and Discussions

### 3.1. Macroscopic Morphology Analysis of Biocomposites

Figure 2 shows photos of (Figure 2a) BC membrane (before purification), (Figure 2b) BC membrane (after purification), (Figure 2c) BC(GO) membrane, (Figure 2d) Ag-BC(GO) membrane, (Figure 2e) PPy-BC(GO) membrane, and (Figure 2f) Ag-PPy/BC(GO) membrane, respectively. By observing Figure 2b,c, it can be observed that BC appears transparent and colorless, while BC(GO) exhibits distinct dispersed dark particles, indicating the random distribution of GO within the BC matrix [32]. Figure 2d,e shows the uniform loading of PPy and Ag within the BC(GO) matrix. Pyrrole, originally the colorless liquid, was successfully mixed with BC(GO) to form the black PPy-BC(GO) after oxidation with FeCl_3_. The surface of Ag-PPy/BC(GO), as seen in Figure 2f, becomes rough due to physical drying, and the film appears black with numerous visible silver-white tiny spots on the surface.

### 3.2. Analysis of Functional Group Structure of Biocomposites

Figure 3 displays the Fourier-transform infrared spectroscopy (FTIR) spectra of the BC membrane (Figure 3a), BC(GO) membrane (Figure 3b), Ag-BC(GO) membrane (Figure 3c), PPy-BC(GO) membrane (Figure 3d), and Ag-PPy/BC(GO) membrane (Figure 3e), respectively. The absorption peaks at 3454 cm^−1^ and 1212–780 cm^−1^ correspond to the asymmetric stretching vibrations of hydroxyl groups (–OH) and the carbon-oxygen bonds (C–O-C) in BC, respectively. The addition of additives does not affect the characteristic functional groups of BC. With the incorporation of GO, the intensity of the hydrocarbon peak decreases, indicating that some hydrocarbon groups on the surface of BC are replaced by the functional groups from GO.

The presence of the absorption peaks at 1701 cm^−1^ (carbonyl group, C=O) and 1063 cm^−1^ (C–O symmetric stretching) confirm the successful oxidation of graphene into GO. Additionally, the peaks at 1074 cm^−1^ and 1540 cm^−1^ are attributed to the stretching vibrations of C–N and C–C in the pyrrole ring, respectively. Notably, the C–C stretching band exhibits a weaker intensity. These characteristic peaks signify the successful integration of PPy onto the BC(GO) membrane. Furthermore, the peak at 2339 cm^−1^ corresponding to the O-H stretching vibration exhibited reduced intensity and a blue shift [33], indicating interactions between silver ions and hydroxyl groups in the composite membrane.

### 3.3. Analysis of Microstructure and Element Distribution of Biocomposites Figures, Tables and Schemes

SEM images in Figure 4 show the morphologies of BC(GO) (Figure 4a,a′), PPy-BC(GO) (Figure 4b,b′), Ag-BC(GO) (Figure 4c,c′), and Ag-PPy/BC(GO) (Figure 4d,d′) composite membranes, respectively. As shown in Figure 4a,a’, BC(GO) exhibits a disordered structure, where the introduction of GO alters the configuration of the BC nanomatrix, resulting in an interconnected network with no visible pores [34,35]. Upon the incorporation of PPy, as observed in Figure 4b,b′, PPy uniformly polymerizes on the BC(GO) surface, transforming it from a smooth and poreless structure to a dense porous network. This porous structure provides ample space for charge exchange between the PPy matrix and electrolyte [36].

Figure 4c,c′ reveals that BC(GO) undergoes impregnation with AgNO_3_ and subsequent UV reduction, resulting in the random attachment of AgNPs onto BC(GO) [5]. Based on the scale bar estimation, the size of these AgNPs is approximately 10~20 nm. Figure 4d,d′ demonstrates the presence of a rough surface on Ag-PPy/BC(GO) membranes, which is characterized by numerous AgNPs. Interestingly, the membrane surface also displays prominent large-sized particles, exceeding sizes of over a hundred nanometers. The membrane surface also displays prominent large-sized particles, exceeding sizes of over a hundred nanometers. By integrating the EDS delamination image of the Ag-PPy/BC(GO) membrane (see Figure 5) with literature analysis, we deduce that these conspicuous large particles are likely a consequence of the ongoing growth of AgNPs [37], thereby contributing to the formation of enlarged silver particles.

Figure 5 shows the elemental distribution on the surfaces of PPy-BC(GO), Ag-BC(GO), and Ag-PPy/BC(GO), respectively. Seen from Figure 5, it is evident that PPy-BC(GO) prominently contains iron, nitrogen, and carbon atoms, which can be attributed to the involvement of Fe^3+^ as an oxidizing agent during the reduction of Pyrrole to PPy. Ag-BC(GO) exhibits the presence of silver, nitrogen, and carbon atoms, with silver elements uniformly distributed on the surface of BC(GO). In Ag-PPy/BC(GO), the existence of silver, nitrogen, and carbon is clearly observed. The silver element originates not only from the AgNPs but also from some larger-sized silver particles aggregated on the membrane surface. This confirms the successful deposition of antibacterial particles onto the BC(GO) matrix.

### 3.4. Swelling Performance Analysis of Biocomposites

Figure 6 presents the rehydration ratio of the BC membrane, BC(GO) membrane, and the different formulations of Ag-PPy/BC(GO) composite membranes. As shown in Figure 6, both the pure BC membrane and the BC(GO) membrane exhibit high swelling ratios, indicating good water absorption properties. In order to better understand the roles of Ag and PPy in biocomposites, composite films were prepared using different concentrations of Ag and Py. A1P1, A1P2, and A1P3 signified the Ag-PPy/BC(GO) films prepared using a fixed Ag concentration (10 g/L) and varying Py concentrations (5 mL/L, 10 mL/L, and 15 mL/L). A2P1, A2P2, and A2P3 represented the Ag-PPy/BC(GO) films synthesized using a constant Ag concentration (15 g/L) and different Py concentrations (5 mL/L, 10 mL/L, and 15 mL/L). A3P1, A3P2, and A3P3 indicated the Ag-PPy/BC(GO) films fabricated with a consistent Ag concentration (20 g/L) and diverse Py concentrations (5 mL/L, 10 mL/L, and 15 mL/L).

However, upon the addition of PPy and Ag, the swelling ratio of the composite membranes significantly decreases from around 80% ± 2% to approximately 34% ± 2%. This reduction could be attributed to the surface polymerization of PPy within the membranes, leading to a decrease in free spaces within the membranes’ structure [38]. As a result, the distribution of water within the composite membrane’s network structure is hindered to some extent, leading to a decline in its water-holding capacity.

Additionally, it is noteworthy that at an Ag concentration (the concentration of silver added during the reaction preparation) of 10 g/L (A1P1, A1P2, A1P3), the rehydration ratio of the composite membrane is slightly higher than that at an Ag concentration of 15 g/L (A2P1, A2P2, A2P3) and 20 g/L (A3P1, A3P2, A3P3) (*p* < 0.05). This suggests that increasing the Ag concentration in the biocomposite membrane may further reduce its rehydration ratio. However, when the Ag concentration continues to increase (e.g., at 15 g/L and 20 g/L), the rehydration ratio of the biocomposite membrane remains relatively stable, with minimal variation ranging from 29.3% to 33.6%.

### 3.5. Analysis of Mechanical Properties of Biocomposites

Figure 7 presents the tensile strength and elongation at the break of the biocomposites. As shown in Figure 7, the pure BC membrane exhibits the highest tensile strength (23.2 MPa), while the addition of GO, Ag, and PPy significantly reduces the tensile strength. This could be attributed to the aggregation of particles on the surface of the composite membrane after Ag and PPy treatment (see Figure 4), leading to a decrease in intermolecular forces which consequently reduces the tensile strength [37].

Regarding the elongation at the break of the biocomposites, the addition of GO and Ag significantly increases the elongation at the break of BC(GO) and Ag-BC(GO) membranes (from the original 12.8% to 28.0% and 32.9%, respectively). However, the elongation at the break slightly decreases to 10.9% and 11.8% for PPy-BC(GO) and Ag/PPy-BC(GO) membranes, respectively, indicating that the addition of PPy has minimal influence on the elongation at the break. Overall, the mechanical properties of the biocomposites decrease with the increase of additives.

### 3.6. Antibacterial Effects of Biocomposites

Figure 8 presents the growth inhibition curves of *E. coli* for the BC(GO) membrane (Figure 8a), PPy-BC(GO) membrane (Figure 8b), Ag-BC(GO) membrane (Figure 8c), and Ag-PPy/BC(GO) membrane (Figure 8d), respectively. Based on the results depicted in Figure 8a, it can be observed that the introduction of GO to the BC membrane BC(GO) and the incorporation of varying concentrations of PPy into the BC(GO) membrane (P1, P2, and P3 corresponded to the PPy-BC(GO) films fabricated at Py concentrations of 5 mL/L, 10 mL/L, and 15 mL/L, respectively) do not demonstrate any inhibitory effects on the OD500 of *E. coli*. Instead, these composite membranes show a promoting effect on *E. coli* growth, indicating that the addition of PPy alone to the BC(GO) membrane is insufficient in inhibiting the growth and proliferation of *E. coli*. This observation contrasts with existing literature reports, which suggest that PPy possesses an antibacterial capability [23,39]. This discrepancy may be attributed to PPy’s strong net negative charge [40], resulting in repulsive forces between the negatively charged *E. coli* and the biocomposite, hindering bacterial adhesion to the PPy-BC(GO) membranes.

Figure 8b illustrates the impact of different concentrations of silver (Ag) in the BC(GO) membrane (A1, A2, and A3 denoted the Ag-BC(GO) films synthesized with Ag concentrations of 10 g/L, 15 g/L, and 20 g/L, respectively) on the growth curve of *E. coli*. Notably, the Ag-BC(GO) membranes, after the incorporation of Ag, exhibit a significant inhibitory effect on the OD500 of *E. coli*, and this inhibitory effect becomes more pronounced with increasing Ag concentrations. Among them, the composite membrane A3 (Ag concentration of 20 g/L) shows the most substantial reduction in bacterial density (OD500) after 48 h of co-culturing with *E. coli*, indicating the most effective antibacterial performance. The growth inhibition rate of *E. coli* after 48 h reaches its highest value at 63%.

Figure 8c displays the 48-h antibacterial growth curves of *E. coli* for Ag-PPy/BC(GO) composite membranes with varying concentrations of Ag and PPy after their incorporation into BC(GO) (refer to Section 3.4 for detailed design of Ag and PPy concentrations). Based on Figure 8c, it is evident that the addition of PPy and Ag results in all Ag-PPy/BC(GO) composite membranes exhibiting significantly enhanced inhibitory capabilities against *E. coli* compared to Ag-BC(GO) membranes (as shown in Figure 8b). Considering that the PPy-BC(GO) membranes do not exhibit growth inhibition of *E. coli* (as observed in Figure 8a), this suggests a synergistic enhancement of antibacterial effects between PPy and Ag [39]. The variations in Ag-PPy/BC(GO) membranes with different Ag and PPy concentrations do not result in substantial changes in the 48-h growth inhibition rates of *E. co*li, which remain within the range of 75% to 84% (as depicted in Figure 8d). Overall, the highest Ag concentration (A3P1, A3P2, A3P3) consistently exhibits the highest and most stable inhibitory rates of *E. coli* growth at 24-h and 48-h, with a range of 83% to 84%, indicating the critical role of Ag concentration as a key determinant of the antibacterial efficacy in Ag-PPy/BC(GO) composite membranes.

It is worth noting that at the highest PPy concentration (15 mg/L), the 48-h growth inhibition rate of *E. coli* is lower for A1P3 than for A1P1 and A1P2, and similarly, for A2P3 compared to A2P1 and A2P2, under the same Ag concentration conditions. This implies that continuously increasing PPy concentration does not necessarily result in a proportional enhancement of growth inhibition against gram-negative bacteria.

Figure 9 illustrates the antibacterial growth curves and growth inhibition rates (24-h and 48-h) of *Staphylococcus aureus* (*S. aureus*) for Ag-PPy/BC(GO) composite membranes. Comparing different Ag and PPy concentration combinations, the 48-h growth inhibition rates of *S. aureus* range from 73% to 84% (as shown in Figure 9a). Overall, the highest growth inhibition rates for *S. aureus* at 24-h and 48-h are observed for A3P3 (Ag concentration of 20 g/L, PPy concentration of 15 mg/L), reaching 88% and 84%, respectively. At an Ag concentration of 10 g/L (A1P1, A1P2, A1P3), the Ag-PPy/BC(GO) composite membranes exhibit a relatively high and stable growth inhibition rate of *S. aureus* at 48-h, ranging from 82% to 84%. This may be attributed to the sufficient release of Ag ions into the solution at lower silver concentrations, where increasing the PPy concentration did not significantly alter the antibacterial efficacy.

Notably, in Figure 9b, the growth inhibition rates of *S. aureus* for Ag-PPy/BC(GO) composite membranes followed the order A2P3 > A2P2 > A2P1 and A3P3 > A3P2 > A3P1. This suggests that, under unchanged Ag concentration conditions, increasing PPy concentration effectively enhances the inhibitory effects of Ag-PPy/BC(GO) composite membranes on *S. aureus*. This might be due to an increased surface roughness or specific surface area of the composite film with higher PPy concentration, facilitating more contact between *S. aureus* and released Ag ions, thereby improving the antibacterial effect. This observation contrasts with the earlier analysis, where increasing PPy concentration did not necessarily result in a proportional enhancement of growth inhibition against gram-negative bacteria. These findings emphasize the importance of optimizing the concentration ratios of primary components in the formulation design of Ag-PPy/BC(GO) composite membranes to achieve the best antibacterial performance.

## 4. Conclusions

This study presents a method for preparing Ag-PPy/BC(GO) biocomposites with high antibacterial activity. Firstly, BC was obtained through the fermentation of Xinhui citrus juice. GO was then in situ composited with BC to introduce a significant number of active oxygen-containing groups onto the BC surface. Subsequently, PPy modification was employed to create a dense porous network structure in BC(GO), enhancing its biocompatibility. Finally, AgNPs were loaded onto PPy/BC(GO) via an impregnation method to confer antimicrobial properties on the material. 

The tensile strength of the Ag-PPy/BC(GO) membranes decreases compared to pure BC membranes, while the elongation at break remains comparable. The combined effect of PPy and AgNPs in the Ag-PPy/BC(GO) membranes exhibit synergistic antibacterial activity, and the optimization of the concentration ratios of the primary components in the formulation design of Ag-PPy/BC(GO) membranes is crucial for achieving the best antibacterial performance. In this study, Ag-PPy/BC(GO) membranes achieved a maximum 48-h growth inhibition rate of 84% against both *E. coli* and *S. aureus.* These findings highlight the promising potential of Ag-PPy/BC(GO) membranes as antibacterial materials. Moreover, the proposed biocomposite membranes demonstrate significant potential for the packaging and preservation of diverse products, such as food and agricultural products.

## Figures and Tables

**Figure 1 micromachines-14-01809-f001:**
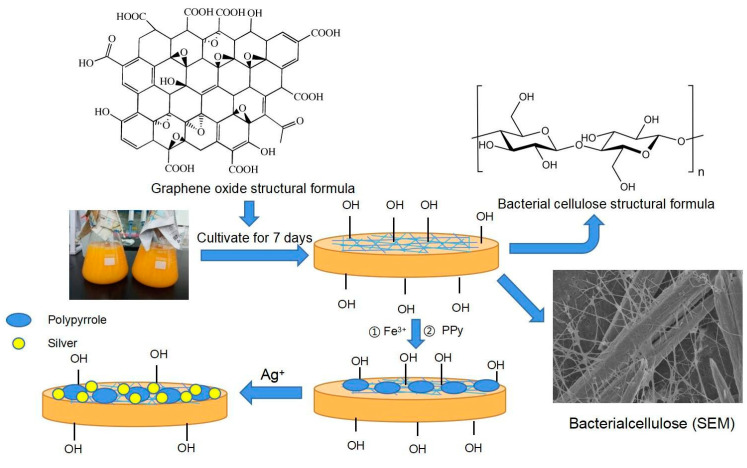
Schematic illustration of the preparation of Ag-PPy/BC(GO).

**Figure 2 micromachines-14-01809-f002:**
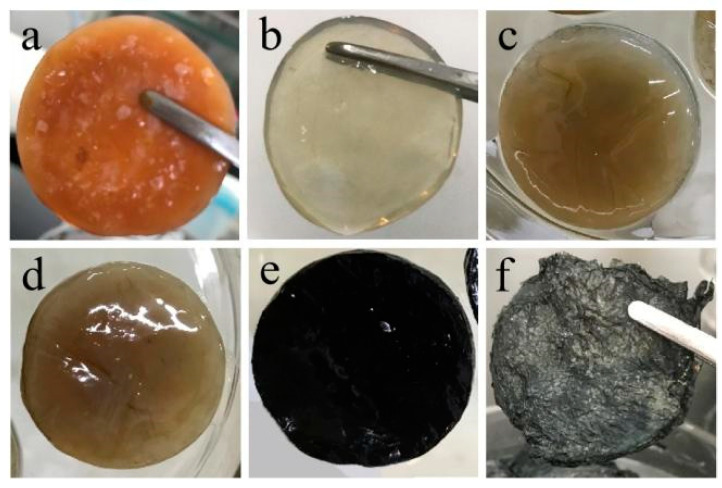
Macro diagram of the laminated film: (**a**) BC (before purification); (**b**) BC (after purification); (**c**) BC(GO); (**d**) Ag-BC(GO); (**e**) PPy-BC(GO); (**f**) Ag-PPy/BC(GO).

**Figure 3 micromachines-14-01809-f003:**
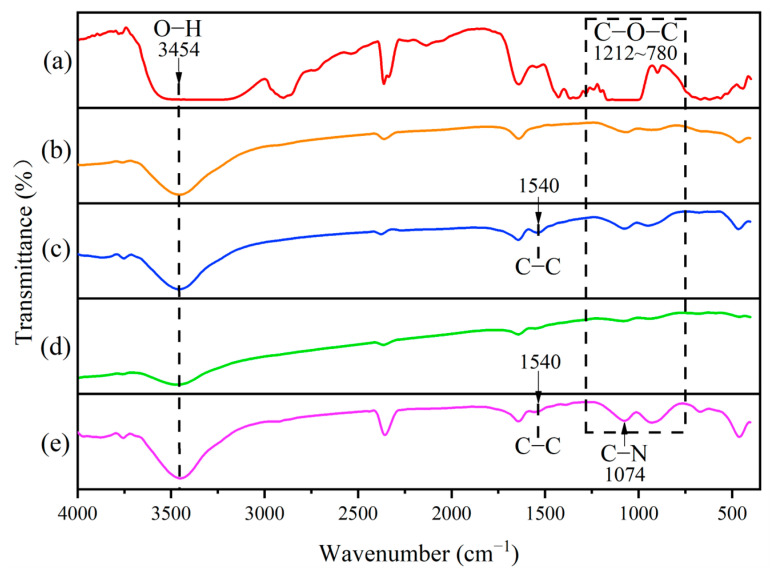
FTIR spectra of: (**a**) BC, (**b**) BC(GO), (**c**) PPy-BC(GO), (**d**) Ag-BC(GO), (**e**) Ag-PPy/BC(GO).

**Figure 4 micromachines-14-01809-f004:**
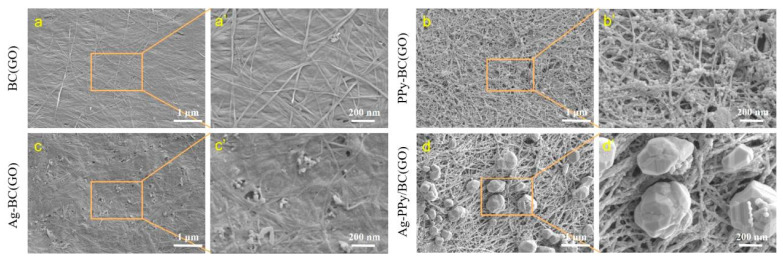
SEM images of samples at different magnifications: (**a**–**d**) (1 μm); (**a**′–**d**′) (200 nm).

**Figure 5 micromachines-14-01809-f005:**
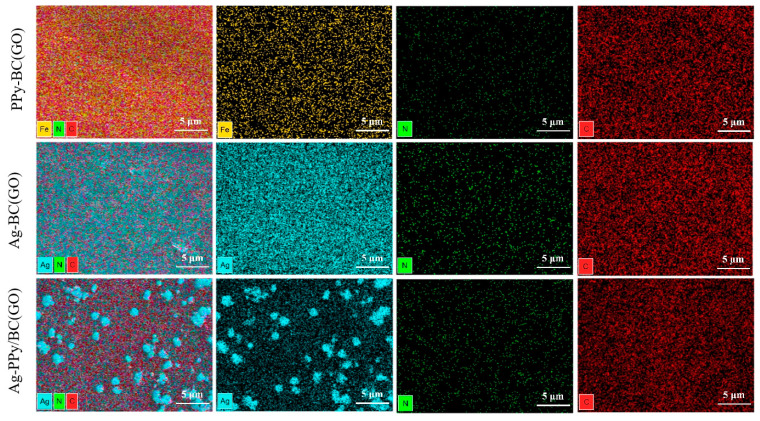
EDS delamination images of the composite film surface.

**Figure 6 micromachines-14-01809-f006:**
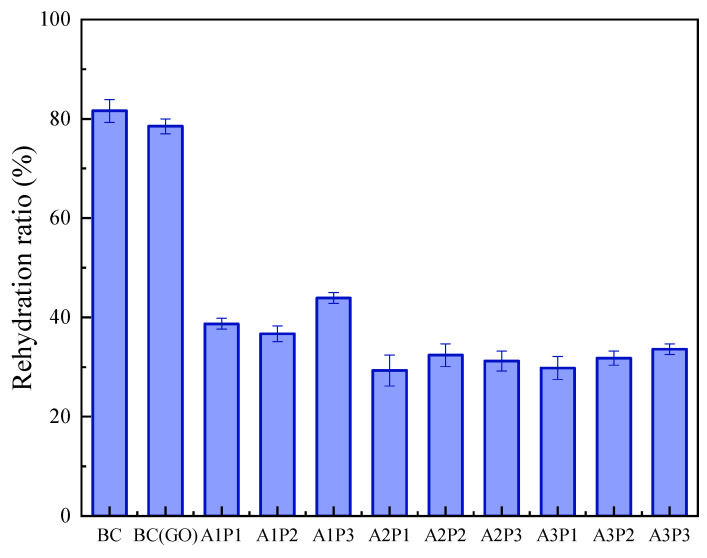
Rehydration ratios of biocomposites.

**Figure 7 micromachines-14-01809-f007:**
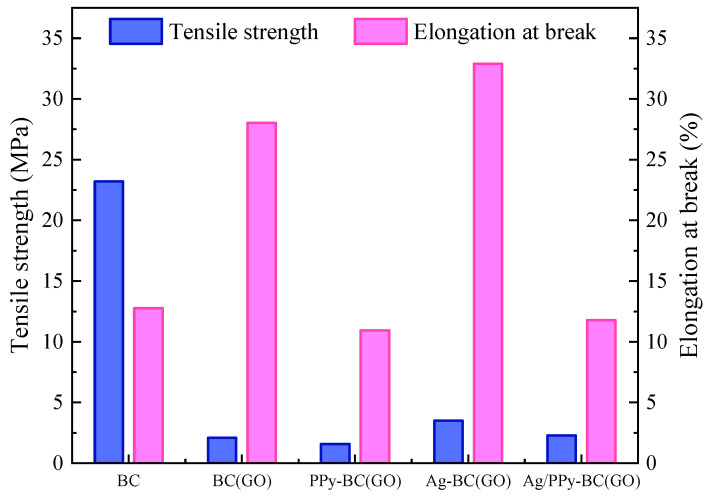
Determination of mechanical properties of biocomposites.

**Figure 8 micromachines-14-01809-f008:**
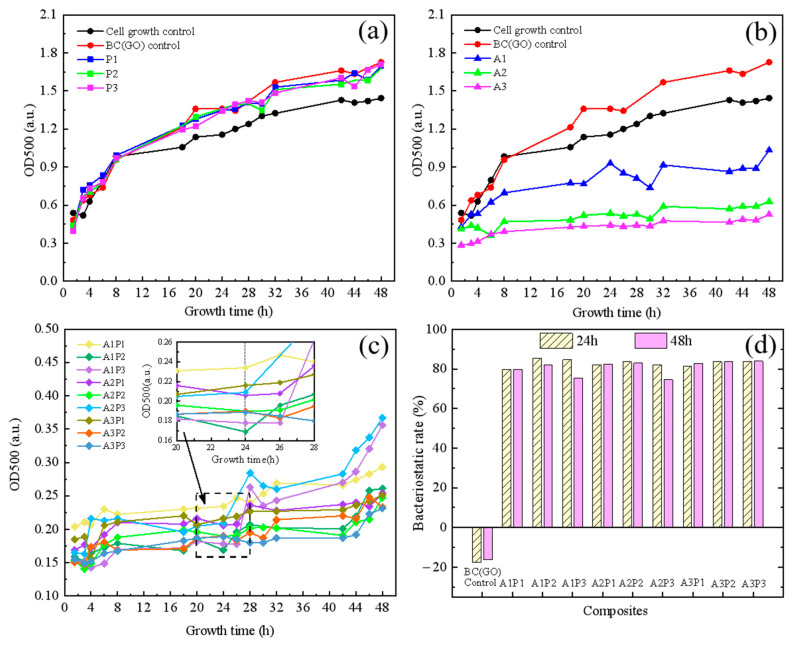
*E.coli* bacteriostatic test result chart: (**a**) bacteriostasis growth curves of PPy; (**b**) bacteriostasis growth curves of nano-silver; (**c**) bacteriostasis growth curves of composite membranes; (**d**) the bacteriostatic rates of composite membranes at 24-h and 48-h.

**Figure 9 micromachines-14-01809-f009:**
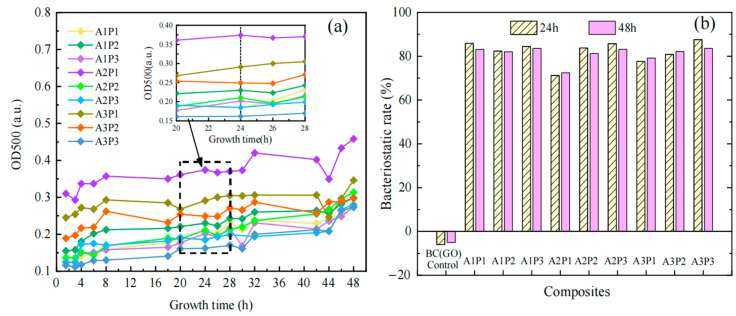
(**a**) *S. aureus* bacteriostasis growth curves of composite membranes; (**b**) the bacteriostatic rates of composite membranes against *S. aureus* at 24-h and 48-h.

## Data Availability

The necessary data availability statements are provided in the appropriate section of the article.

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
