# Peer review of "Nano Ag/PPy Biocomposites Based on Graphene Oxide Modified Bacterial Cellulose from the Juice of Xinhui Citrus and Its Antibacterial Activity"

_micromachines, 2023, doi:10.3390/mi14101809_

Round 1
Reviewer 1 Report
The purpose of the work should be clearly formulated.
line 73-75......This research contributes to a deeper understanding of the interactions between GO, PPy, AgNPs, and bacteria.....but in the article many phenomena are described, but their cause is not explained.
The procedure for obtaining the composite membranes is poorly described.
It is not explained why FeCl3 aqueous solution is used to prepare a composite membrane. Then it is said that (paragraph 2.2.3.)... The dried membrane was immersed in a certain concentration of AgNO3 solution, but it is not clear what happens to the Fe ions in the membrane, do they remain or are they removed?
It is not clear where exactly colored membranes can be used.
The article talks about silver nanoparticles, but does not provide the results of direct measurements of the size of silver particles.
The article says... (line 230-232) Figure 4 d and 4d' demonstrate that the surface of Ag-PPy/BC(GO) appears rough with numerous particle aggregates, which may be attributed to the reaction between residual Cl- and some Ag+ forming AgCl precipitates.
Perhaps residual amounts of iron ions also affect the properties of composite membranes?
The article does not contain information about the possible use of composite membranes with antibacterial properties: food films, packaging for agricultural products, medical products or something else?
The conclusions state that ......The mechanical properties of the Ag-PPy/BC(GO) membranes decrease with the increase of Ag and PPy. ...In this study, Ag-PPy/BC(GO) membranes achieved a maximum 48-h growth inhibition rate of 84% against both E. coli and S. aureus.
It is not clear what mechanical characteristics are achieved.
Reviewer 2 Report
This article uses waste citrus fruit gravy after fermentation as raw material to prepare BC membrane, and introduces GO, PPy and Ag NPs materials into BC, expanding its potential in antibacterial applications. However, there are some problems that the authors should notice. I suggested its acceptance for publication after some revision.
1. In the part of 3.2, the description of evidence for the successful synthesis of PPy onto BC (GO) membranes is repeated.
2. In the part of 3.5, the punctuation in this section is incomplete, such as: “ ) ”. The similar questions should be checked in the manuscript.
3. For the interpretation of Figure 9, the concentration of Ag in A3P3 (30g/L) is not the same as the previous article (20g/L). Such similar questions should be checked.
4. For the Figure 9b, authors only list the positive correlation between the concentration of PPy in the composite membrane and the growth inhibition of Staphylococcus aureus at A2 and A3 (Ag concentration 15g/L and 20g/L), the abnormal phenomenon that the concentration of PPy in the composite membrane was not positively correlated with the growth inhibition of Staphylococcus aureus at A1 (Ag concentration 10g/L), why?
Minor editing of English language required.
